# New Insights into the Functions of MicroRNAs in Cardiac Fibrosis: From Mechanisms to Therapeutic Strategies

**DOI:** 10.3390/genes13081390

**Published:** 2022-08-04

**Authors:** Yuanyuan Zhao, Dunfeng Du, Shanshan Chen, Zhishui Chen, Jiajia Zhao

**Affiliations:** 1Institute of Organ Transplantation, Tongji Hospital, Tongji Medical College, Huazhong University of Science and Technology, Wuhan 430030, China; 2Key Laboratory of Organ Transplantation, Ministry of Education, and NHC Key Laboratory of Organ Transplantation, Wuhan 430030, China; 3Key Laboratory of Organ Transplantation, Chinese Academy of Medical Sciences, Wuhan 430030, China; 4Key Laboratory for Molecular Diagnosis of Hubei Province, The Central Hospital of Wuhan, Tongji Medical College, Huazhong University of Science and Technology, Wuhan 430070, China; 5Department of Stomatology, Union Hospital, Tongji Medical College, Huazhong University of Science and Technology, Wuhan 430022, China

**Keywords:** cardiac fibrosis, miRNAs, cardiac fibrosis-related signaling pathways, clinical application

## Abstract

Cardiac fibrosis is a significant global health problem associated with almost all types of heart disease. Extensive cardiac fibrosis reduces tissue compliance and contributes to adverse outcomes, such as cardiomyocyte hypertrophy, cardiomyocyte apoptosis, and even heart failure. It is mainly associated with pathological myocardial remodeling, characterized by the excessive deposition of extracellular matrix (ECM) proteins in cardiac parenchymal tissues. In recent years, a growing body of evidence demonstrated that microRNAs (miRNAs) have a crucial role in the pathological development of cardiac fibrosis. More than sixty miRNAs have been associated with the progression of cardiac fibrosis. In this review, we summarized potential miRNAs and miRNAs-related regulatory mechanisms for cardiac fibrosis and discussed the potential clinical application of miRNAs in cardiac fibrosis.

## 1. Introduction

Cardiac fibrosis is one of the most important global health problems, significantly associated with almost all types of heart disease. It mainly involves pathological myocardial remodeling characterized by the excessive deposition of extracellular matrix (ECM) proteins in cardiac parenchymal tissues [1]. Cardiac fibrosis typically has a protective role in the replacement of damaged or dead cardiomyocytes through the formation of a scar when the heart suffers from several pathophysiological conditions, such as acute myocardial infarction (MI), pressure and volume overload, aging, etc. Although the scar helps maintain the structural integrity of the chamber, it also decreases the contractile capacity of cardiomyocytes [2]. In 2019, the Committee of Translational Research of the Heart Failure Association (HFA) of the European Society of Cardiology classified the fibrotic process in the heart into “reparative/replacement” and “reactive” fibrosis [3]. The “reparative/replacement” type replaces myocardial areas where cardiomyocytes underwent cell death, i.e., ischemic events, and is common after myocardial infarction (MI), while “reactive” fibrosis is driven by a series of stimuli, e.g., inflammatory cells, paracrine signals, and oxidative stress [3]. In fact, many features are shared between different types of fibrosis. For example, upon systemic pressure overload, fibrosis in the heart might diffuse and react to the mechanical changes that are occurring. With acute injury such as MI, fibrosis might initially have reparative functions, leading to a focal scar that replaces dying cardiomyocytes. Extensive cardiac fibrosis reduces tissue compliance and contributes to adverse outcomes, including cardiomyocyte hypertrophy, cardiomyocyte apoptosis, and even heart failure.

In cardiac fibrosis, activated fibroblasts and myofibroblasts are the two central effector cells, serving as the main source of ECM proteins [4]. Fibroblasts are cells of mesenchymal origin that produce structural ECM proteins. Under injury or stress, they are transformed into myofibroblasts [5]. Transdifferentiation of fibroblasts into myofibroblasts is the key cellular event for a fibrotic response. In addition, several types of cells, such as immune cells, cardiomyocytes, and vascular cells, can also acquire a fibrogenic phenotype under certain conditions [5].

Additionally, many molecular pathways have been demonstrated to exert important roles in cardiac fibrosis. Inflammatory signals involving intense activation of cytokine and chemokine cascades have an important function during ischemic and reparative fibrosis [6]. Meanwhile, the angiotensin/aldosterone axis and fibrogenic growth factors, such as tumor necrosis factor-α (TNF-α), interleukin (IL)-1, IL-10, chemokines, transforming growth factor-β (TGF-β), IL-11, and platelet-derived growth factors (PDGFs), participate in fibrotic cardiac conditions regardless of etiology [7]. Thus, understanding pathogenetic mechanisms and risk factors of cardiac fibrosis are essential for developing therapeutic and preventive strategies.

In recent years, increasing studies have shown that non-coding RNAs (mainly including microRNAs (miRNAs), circular RNAs (circRNAs) and long non-coding RNAs, competing for endogenous RNAs (ceRNAs), etc.) have a crucial role in the pathological development of cardiac fibrosis. Over the past twenty years, great progress has been made in researching the role of non-coding RNAs, particularly miRNAs, in multiple diseases. miRNAs are small RNAs with a length of approximately 19–25 nucleotides encoded by both eukaryotes and viruses. miRNAs are involved in Watson–Crick pairing with miRNA response elements, which are typically located in the 3′ untranslated region (UTR) of mRNAs to guide Argonaute (AGO) proteins to their target RNA transcripts [8]. Furthermore, miRNAs mainly bind to messenger RNAs (mRNAs) in a complementary manner and induce gene silencing by inhibiting the translation or degradation of mRNAs [9]. Most miRNAs are present in the nucleus and the cytoplasm and can be released into the circulating system, and their expression is controlled by epigenetic modifications in a tissue-specific manner [10]. Recently, more than 60 miRNAs have been reported to be associated with cardiac fibrosis. In this review, we summarized potential miRNAs and miRNAs-related regulatory mechanisms for cardiac fibrosis and discussed the therapeutic strategies of cardiac fibrosis through the modulation of miRNAs.

## 2. The Pathophysiology of Cardiac Fibrosis

### 2.1. Origin and Maintenance

The activated fibroblasts, the main ECM-producing cells, are widely accepted as the central cellular effectors of myocardial fibrosis. Cardiac fibroblasts are typically involved in the perimysium and endomysium in the mammalian heart [11]. Herein, we mainly discussed the “reparative/replacement” fibrosis.

In normal cardiac tissue, cardiac fibroblasts maintain the structural integrity of the matrix network and regulate collagen turnover [4]. Cardiac fibrosis is mediated by the activation of resident cardiac fibroblasts, which differentiate into myofibroblasts in response to injury or stress. Subsequently, the apoptosis of myofibroblasts may cause rapid scar formation in the damaged tissues. Therefore, activated fibroblasts and myofibroblasts are very important for local wound healing response through the expansion and remodeling of ECM. In vivo and in vitro studies have demonstrated that the fibroblasts are activated by the TGF-β1/Smad signaling pathway [12,13,14]. TGF-β1 has an important role in mediating cell growth and differentiation, wound repair, and ECM production [15], while the nuclear accumulation of active Smad complexes is crucial for the transduction of TGF-β-superfamily signals from transmembrane receptors into the nucleus [16]. In addition, the Hippo signaling pathway has been confirmed as a novel fibroblast-activating pathway (Figure 1) [16]. A recent investigation revealed that the Hippo signaling pathway typically maintains cardiac fibroblasts in the resting state. When a YAP-induced injury response switches off the pathway, cardiac fibroblasts are activated [17].

### 2.2. Outcomes

Although cardiac fibrosis is a hallmark of pathological conditions after heart injury, sustained fibrotic responses lead to distorted heart construction and cardiac dysfunction and increase the risk of various types of heart disease. First, cardiac fibrosis is one of the major factors leading to cardiac remodeling that impairs mechano-electric coupling, thereby contributing to arrhythmias [18]. Cardiac fibrosis leads to cardiac structural changes, including increasing myocardial stiffness and impairs relaxation and contractile force [19]. On the other hand, ECM and collagen deposition may reduce the electrical coupling between myocytes and facilitate focal activities [20]. Second, myofibroblasts can secrete growth factors, which directly induce cardiac hypertrophy and further promote heart failure outcomes [21]. In addition, cardiac fibrosis affects the heart by accelerating the remodeling process [7].

## 3. The Functions of miRNAs in Cardiac Fibrosis-Related Signaling Pathways

According to our literature review, more than 60 miRNAs are directly or indirectly involved in anti- or pro-fibrosis in the cardiac tissues (Table 1). Herein, we reviewed and discussed the functions of miRNAs in cardiac fibrosis-related signaling pathways.

### 3.1. TGF-β Signaling Pathway

The TGF-β signaling pathway has an important role in cardiac fibrosis. TGF-β can induce signal transduction via Smad-dependent (canonical) and Smad-independent (non-canonical) pathways. A previous study demonstrated that cardiac-restricted expression of a mutant TGF-β1, inhibiting covalent tethering of the TGF-β latent complex to the ECM, is associated with atrial rather than ventricular fibrosis, suggesting the increased susceptibility of the atrial myocardium to the fibrogenic actions of TGF-β [72]. Among them, Smad-dependent signaling cascades are more important in regulating myocardial remodeling and fibrosis than Smad-independent pathways [73]. Numerous studies confirmed that TGF-β can induce target genes to promote the growth and viability of fibroblasts. In mammals, there are three isoforms of TGF-β including TGF-β1 TGF-β2, and TGF-β3. In the heart, the progression of fibrosis is mainly attributed to TGF-β1 [74]. This isoform has been reported as a prototypical fibrogenic cytokine with high expression levels in human and animal models of cardiac fibrosis [75].

Recently, numerous miRNAs have been demonstrated to have a key role in cardiac fibrosis via the TGF-β signaling pathway. In the canonical TGF-β signaling pathway (Figure 2), TGF-β1 interacts with *TGFβRI/TGFβRII*, leading to the phosphorylation of *Smad2/3*. Studies demonstrated that miR-101a [23], miR-675 [51], and miR-15 [29] might act as anti-fibrotic molecules by suppressing *TGFβRI*. Similarly, miR-9 [52], miR-590 [25], and miR-145 [28] were reported to have an anti-fibrotic role by suppressing *TGFβRII*. MiR-27b can reduce expression levels of α-SMA and collagen proteins by targeting *ALK5* and blocking the Smad-2/3 signaling pathway [39], while miR-223 can enhance the expression levels of α-SMA and collagen proteins [62]. In contrast, miR-10a promotes cardiac fibrosis by increasing the expression levels of α-SMA, collagen-I, and collagen [53], while miR-155 leads to hyperactivation of pro-fibrotic TGF-β1/Smad3 signaling pathway and the excessive cardiac fibrosis [56]. Additionally, miR-133 [25], miR-633 [50], miR-425 and miR-744 [48], and miR-22 [33] can directly target *TGF-β1* and reduce its expression level. However, some studies reported that miR-22 could also target *TGFβR1* and *PTAFR* to achieve the anti-fibrotic capacity [34,35], suggesting that miRNAs could regulate the pathway by simultaneously targeting multiple crucial genes. In addition, the regulation of TGF-β1 can occur at some other levels. For instance, a previous study found that miR-34a expression is upregulated in cardiac tissue after MI and can reduce the activity of TGF-β1 by directly targeting *Smad4* [67]. Inhibition of miR34a decreased cardiac fibrosis in mice, which suggested that the TGF-β1 signaling pathway might be a central target for cardiac fibrosis therapy. Furthermore, mir-410-5p can inhibit *Smad7* expression level, activating the phosphorylation of *Smad2* and cardiac fibrosis [69]. In addition, Wang et al. found that miR-24 can target *FURIN* (a protease that controls latent TGF-β activation) to suppress the TGF-β secretion and Smad2/3 phosphorylation in cardiac fibroblasts [36]. A few years later, Tao et al. found that miR-433 could induce TGF-β1 expression level by inhibiting *AZIN1* expression level [70], which is a regulator of polyamine synthesis with an undefined relationship with TGF-β1 Furthermore, after MI, the miR-21 and miR-328 decrease TGFβRIII expression level, which could directly neutralize TGF-β1 and contribute to excessive ECM production and cardiac fibrosis [57,66].

In non-canonical responses, three pathways (PI3K/Akt, RhoA-ROCK axis, and MAPK cascades) can be directly activated by TGF-β [33]. A few studies reported that TGF-β1 regulated cardiac fibrosis through the activation of the non-canonical responses [76]. Tao et al. demonstrated that miRNA-29a could act as a key regulator to suppress the proliferation of cardiac fibroblasts by targeting the VEGF-A/MAPK signaling pathway [41]. Yet, more evidence about the regulation of non-canonical TGF-β responses by miRNAs is needed. The above-mentioned findings indicate that the TGF-β signaling pathway could promote cardiac fibrosis mainly through the Smad-dependent pathway.

In addition, the TGF-β signaling pathway has a critical role in tissue repair, remodeling, and regeneration. Therefore, TGF-β has been suggested as an attractive therapeutic target for patients with MI and cardiac fibrosis [77,78]. However, because activation of TGF-β signaling exerts protective and detrimental effects, the therapeutic translation of TGF-β signaling presents a huge challenge. Thus, attempts for clinical treatment of cardiac fibrosis by targeting TGF-β signaling need to focus on brief therapeutic interventions [77]. We discuss the potential clinical application of miRNAs in the next chapter.

### 3.2. TGF-β–Related Wnt and NF-κB Signaling Pathways

The Wnt signaling pathway is essential for normal development (generation of a normally patterned embryo) and is also involved in many pathogeneses. Similar to the TGF-β signaling pathway, the Wnt signaling pathway includes β-catenin-dependent (canonical) and -independent (non-canonical) cascades. The canonical β-catenin-dependent pathway is involved in the proliferation and differentiation of myofibroblasts and may cause fibrogenesis [79]. Recently, some scholars found an extensive crosstalk between the TGF-β and Wnt signaling pathways [33]. TGF-β can induce Wnt expression through the TGF-β-activated kinase 1 (TAK) signaling pathway and activate the canonical Wnt signaling pathway. This process also leads to the activation and response of nuclear factor kappa-light-chain-enhancer of activated B cells (NF-κB) (Figure 3) [80,81].

miRNAs also have an important role in the crosstalk mentioned above. For instance, miR-25-3p can target Dickkopf (DKK), a regulator of Wnt signaling, and activate Smad3 and fibrosis-related gene expression [63]. The study also proved that in cardiac fibrosis, the NF-κB signaling pathway could upregulate the expression level of miR-25-3p, suggesting that the crosstalk between signaling pathways in cardiac fibrosis could be associated with miRNAs. Additionally, miR-214-3p has been reported to reduce *CTRP9* expression level, a cardioprotective cardiokine, and further induce both TGF-β/Smad and Wnt/β-catenin signaling pathways [31]. However, Aurora et al. presented a method where they used homologous recombination to generate a conditional targeted deletion of the mIR214 and found that genetic deletion of miR-214 in mice could lead to a loss of cardiac contractility, increased apoptosis, and excessive fibrosis [32]. Using the miR-214 knockout mice, they found that miR-214 could protect mice against ischemia-reperfusion injury (IRI) by regulating the *NCX1* expression level. Therefore, the role of miR-214 in cardiac fibrosis is still debatable and should be further explored. Furthermore, miR-384-5p showed a suppressive role in cardiac fibrosis by targeting the key receptors of the TGF-β/Wnt transactivation circuit [47].

### 3.3. Renin-Angiotensin-Aldosterone System (RAAS)

Except for TGF-β and TGF-β–related signaling pathways, the RAAS is another important signaling pathway in cardiac fibrosis. Among them, angiotensin II (Ang-II) was reported as the most predominant isoform [7]. It has been demonstrated that AT1 and AT2 are two specific receptors of Ang-II that have opposite roles in cardiac fibrosis. Ang-II/AT1 regulates cardiac fibroblast activation, as well as ECM synthesis and apoptosis, while Ang-II/AT2 inhibits AT1 action [4,82,83]. As illustrated in Figure 4, the Ang-II/AT1 pathway involving cardiac fibrosis activation could be mediated through extracellular signal-regulated kinase (ERK), p38/MAPK, and protein kinase C (PKC) [84]. To date, several studies have reported that Ang-II could regulate a set of miRNAs to modulate cardiac fibrosis. For example, in adult rat cardiac fibroblasts, Ang-II can significantly induce miR-224 expression. SMAD4, SMAD5, cyclin-dependent kinase 9, and early growth response 1/2 are the potential target genes of miR-224, which means that Ang II upregulates miR-224 expression by providing a starting point of cardiac fibrosis after MI [58]. Furthermore, Ang-II can upregulate miR-323a-3p [64], while miR-1 [22], miR-22-3p, and miR-30b-5p [35] could be downregulated by Ang-II. Moreover, Ang-II could regulate lncRNA expression level, having a crucial role in sponging miRNAs, indirectly affecting miRNAs and their target genes. For instance, Ang-II treatment markedly upregulates circHIPK3 expression level, sponging miR-29b-3p and downregulating the expression levels of its target genes (*COLIA1, COL3A1,* and *a-SMA*) [42]. miR-133a is a well-known miRNA that not only targets TGF-β1, but also COLA1 [85].

In addition, miRNAs have a substantial role in regulating Ang-II and RAAS. In vivo silencing of miR-125b could rescue Ang-II-induced cardiac fibrosis [54]. Moreover, overexpression of miR-327 promoted Ang-II-induced differentiation of cardiac fibroblasts into myofibroblasts [65], while let-7i miRNAs exhibited opposite effects [86].

### 3.4. Other Relevant Pathways

Autophagy has a critical role in heart disease. Autophagy is a highly conserved eukaryotic cellular recycling process that removes unnecessary or dysfunctional components through a lysosome-dependent regulated mechanism. A previous study demonstrated that excessive autophagy could promote the development of cardiac fibrosis [87]. Ji et al. found that miR-26a-5p regulates Unc-51 such as autophagy activating kinase 1 (ULK1) expression level and affects autophagy in cardiac fibroblasts [37]. ULK1 kinase activity is essential for starvation-induced autophagy. Furthermore, miR-132-3p and miR-20a-5p were also confirmed to regulate autophagy in heart failure by targeting pro-autophagic transcription factor (FOXO3a) and ATG7/SOD2, respectively [55].

However, the influence of autophagy on remodeling in heart failure and cardiac fibrosis is still controversial [88]. In fact, autophagic flux was used to evaluate the autophagic activity compared with other methods. However, due to the nature of this pathway, autophagic flux analyses often lack accuracy [89]. It might also affect the accuracy of research conclusions on autophagy and heart failure, and fibrosis. Therefore, further studies are needed to provide more evidence.

In addition, Zhao et al. found that immunoglobulin E (IgE) and its high-affinity receptor (FcεR1) can promote ventricular remodeling and myocardial fibrosis [90] by suppressing miR-486a-5p, while miR-486-5p could attenuate IgE-FcεR1-induced collagen expression levels and Ang-II-induced cardiac fibrosis. Importantly, the miR-486-5p expression level was regulated by Smad1 [49]. These findings revealed a complex network of pathways combined with miRNAs that enormously affected cardiac fibrosis. In addition, isoproterenol (ISO) could induce cardiac fibrosis via the Kelch-like ECH-associated protein 1 (Keap1)/nuclear factor erythroid-2-related factor 2 (Nrf2) signaling pathway, and miR-26b could relieve the effect by targeting Keap1 [38].

## 4. Potential Clinical Application of miRNAs in Cardiac Fibrosis

### 4.1. Potential Therapeutic Targets

The function of miRNAs in treating cardiac fibrosis as new targets has attracted the attention of clinicians. The study of the effects of the expression, stability, and function of miRNAs can be useful in developing new therapeutic approaches [91]. As seen above, miRNAs can target several cardiac fibrosis-related signaling pathways, which can also be used as potential therapeutic targets. To date, several animal studies have been conducted on this topic (Table 2). Herein, we explain some of the most important findings.

#### 4.1.1. Anti-Fibrotic miRNAs

To date, several miRNAs, including miR-29, miR-133, miR-30, miR-1, and miR-101a, have been reported as potential therapeutic targets. Among them, a miR-29 family (which includes miR-29a, miR-29b, and miR-29c) has been shown to alleviate ECM remodeling [43]. Previous studies also showed that the MI-regulated miRNAs are members of the miR-29 family, which are down-regulated in the heart region adjacent to the infarct zone [43,92,93]. A great number of fibrosis-related genes were found as the direct target genes of miR-29, such as *COL1A1-3* (collagens), *FBN-1* (fibrillin-1), and *ELN1* (elastin), indicating that miR-29 could be a key regulator of fibrosis. Furthermore, in vivo study suggested that overexpression of miR-29 in fibroblasts can significantly reduce the expression levels of fibrosis-related genes. Additionally, the overexpression of miR-29b in the mouse cardiac tissue could prevent AngII-induced cardiac fibrosis [44]. Li et al. used an ethanolamine (EA)-functionalized poly (glycidyl methacrylate) s (PGMAs)-based gene vector to deliver miR-29b in vivo [94]. This vector can also be used for the delivery of anti-fibrotic miRNAs. Hence, pharmacological inhibitors for preventing the reduction in miR-29 or overexpression of miR-29 could be clinically valuable.

In their study, Matkovich et al. found that miR-133a is downregulated in transverse aortic constriction (TAC) and isoproterenol-induced hypertrophy, while transgenic expression of miR-133a could improve TAC-induced myocardial fibrosis [95]. Rho-specific GDP/GTP exchange factor, Cdc42 (an Rho-family GTPase), WHSC2 (also known as NELF-A), and connective tissue growth factor (CTGF) were identified as the target genes of miR-133 [26,27]. Importantly, if both isoforms (miR-133a-1 and miR-133a-2) are knocked out, the two-fold effect could be expected. Recently, several drugs have been reported to regulate the miR-133 expression level. For instance, a traditional Chinese herbal medicine, namely tanshinone IIA, can upregulate miR-133 expression levels and further activate ERK1/2 in cardiomyocytes [98]. In addition, both ivabradine (a specific heart rate-lowering drug) and β-blocker carvedilol can improve the functions of cardiomyocytes by upregulating miR-133 expression levels after MI [99,100]. However, the therapeutic effects of these three drugs in cardiac fibrosis have not yet been demonstrated. For the treatment of cardiac fibrosis, another selective β-blocker nebivolol can significantly decrease cardiac remodeling and fibrosis by upregulating the expression levels of miR-133a, miR-27a, and miR-29a [40].

Furthermore, studies found that miR-133, miR-30, and miR-18a could inhibit CTGF expression levels and adjust the structural change in the ECM remodeling [30,45]. Kraus et al. co-cultured cortical bone-derived stem cells (CBSCs) and adult mouse cardiac fibroblasts, which were treated with 10 ng TGF-β for 48 h to mimic cardiac injury. They discovered that CBSCs modulate cardiac fibroblast response via miR-18a in the heart after injury [30]. Using stem cells to improve cardiac function via the secretion of anti-fibrotic factors, including miRNAs, is a very valuable therapeutic strategy.

MiR-1 is a member of the subgroup of striated muscle-specific or muscle-enriched miRNAs. A previous study demonstrated that overexpression of miR-1 using AAV9 could improve cardiac function in TAC-induced cardiac remodeling and heart failure [96]. A recent study performed photobiomodulation therapy (PBMT) on rats with MI and found changes in expression levels of miRNAs. The results showed that PBMT could significantly upregulate the expression levels of miR-1, miR-29a, and miR-133 by 155%, 55.17%, and 150%, respectively [101]. Another research demonstrated that PBMT could improve cardiac fibrosis by modulating the gene transcription and miRNA expressions. Moreover, miR-29, miR-133, miR-30, and miR-1 have been widely reported as the therapeutic targets of cardiac fibrosis [78,93,94,95]. Pan et al. reported that adenovirus-mediated overexpression of miR-101a in rats with chronic MI could remarkably improve cardiac function, which confirmed the therapeutic potential of miR-101a [24]. However, future studies are needed to provide more evidence.

#### 4.1.2. Pro-Fibrotic miRNAs

For anti-fibrotic miRNAs, pharmacological inhibitors could be valuable therapeutic targets, preventing their reduction or overexpression. However, for pro-fibrotic miRNAs, inhibition of their expression levels can be an ideal therapeutic approach. MiR-21 and miR-34 have been proven as anti-fibrotic targets. Herein, we discussed the prospects of these miRNAs.

Several studies demonstrated that miR-21 has a key role in fibrosis. During myocardial ischemia/reperfusion and heart failure, the miR-21 expression level is significantly upregulated [59,102]. miR-21 mainly regulates the expression levels of SPRYI and CADM1 and then upregulates collagens and canonical and non-canonical TGF-β signaling pathways [60]. In a recent study, Nonaka et al. used a locked nucleic acid (LNA)-anti-miR-21 inhibitor to treat mice with chronic Chagas disease cardiomyopathy (CCC), finding that silencing of miR-21 could significantly reduce cardiac fibrosis by the activation of ECM degradation [61]. Furthermore, another study used a pig model of heart failure to assess the therapeutic effects of antimiR-21 by intracoronary infusion. The results showed that antimiR-21-treated pigs exhibited reduced cardiac fibrosis at 33 days after IRI [103]. It is valuable progress that provided the first evidence for the therapeutic efficacy of cardiac fibrosis by targeting miRNAs directly in a large animal model. On the other hand, docosahexaenoic acid (DHA) has been reported to delay the progression of myocardial fibrosis. Furthermore, DHA could enhance PECK expression level and reverse Ang-II-induced miR-21 expression level [104].

Lin et al. showed that miR-34 family members (miR-34a, miR-34b, and miR-34c) are upregulated in the heart in response to stress, including MI [97]. Bernardo et al. reported that using an 8-mer LNA oligonucleotide complementary to the seed region of the miR-34 family to inhibit the expression levels of miR-34 family members could prevent pressure overload-induced left ventricular remodeling and improve cardiac fibrosis. Importantly, only 15-mer LNA oligonucleotide complementary to the seed region of the miR-34a did not show a promising therapeutic efficacy [68]. The results proved the therapeutic potential of targeting an entire family. In this study, scholars developed a method to silence the expression levels of miR-34 family members using an s.c.-delivered seed-targeting 8-mer LNA, which is worthy of use to evaluate the therapeutic effects of other pro-fibrotic miRNAs.

### 4.2. Diagnostic and Prognostic Biomarkers

In addition to the above-discussed therapeutic potential of miRNAs, many cardiac fibrosis-related miRNAs could be used as candidate diagnostic and prognostic biomarkers. The outstanding characteristics of miRNAs, such as stability in the circulation, high sensitivity and specificity, and special regulation in tissues and diseases, are some of the advantages when using miRNAs as biomarkers [105]. In plasma, miRNAs are incorporated into microparticles or bind to proteins or high-density lipoproteins [106]. Due to difficulties in the imaging techniques and protein biomarkers to monitor the development of cardiac fibrosis at the early stage, miRNAs possess noticeable advantages in monitoring various diseases. Previous studies confirmed miR-21 as a major regulator of cardiac fibrosis. Furthermore, circulating miR-21 has a key role in the diagnosis and prognosis of cardiac fibrosis [107,108]. Although miR-21 from a peripheral vein (mirR-21-PV) and coronary sinus (miR-21-CS) showed perfect efficacy in diagnosing heart failure, miRNA-21-CS could be a more effective indicator of cardiac function. Additionally, miRNA-21-CS was found efficient in predicting re-hospitalization due to a strong association between miRNA-21-CS and ejection fraction (EF) [109]. However, Zhou et al. reported that serum level of miR-21 was also associated with atrial fibrillation-free survival after catheter ablation [110]. It seems that miR-21 could not be a specific biomarker associated with cardiac fibrosis. We assumed that there was a similar pathological process in these clinical phenotypes. In addition, Stienen et al. measured a set of circulating miRNAs associated with cardiac fibrosis, including miR-1, miR-21, miR-29a, miR-29b, miR-101, miR-122, and miR-133a in 198 patients from the Eplerenone Post-Acute Myocardial Infarction Heart Failure Efficacy and Survival Study (EPHESUS). They found that only miR-133a could be a biomarker independently associated with the decrease in thr carboxy-terminal propeptide of type-I collagen (PICP, (β-value = 6.43, 95% confidence interval (CI): 12.71 to −0.15), which would be correlated with collagen synthesis and myocardial fibrosis [111,112]. Recently, Thottakara et al. reported that the plasma level of miR-4454 was significantly correlated with the extent of cardiac fibrosis detected by late gadolinium enhancement (LGE) on magnetic resonance imaging (MRI) (r = 0.56) and septal wall-thickness (r = 0.38) in patients with hypertrophic cardiomyopathy (HCM) [71].

### 4.3. New Progress in Delivery Approaches of miRNAs

To date, the therapies based on miRNAs are still facing some challenges. Because naked miRNAs are easily degraded and cannot pass through the extracellular matrix, appropriate tools and techniques should be developed for its delivery. For the past few years, several new techniques have been developed. For instance, Nanoscale vectors have been widely used to deliver miRNAs/siRNAs for the treatment of cancers [113,114]. Nanomedicines can accumulate in tumors based on the enhanced permeability and retention (EPR) effect. However, only a few studies reported the use of nanomedicines for cardiac fibrosis. Moreover, exosomes, which are considered as “nanosphere”, have been identified as a useful tool for transporting miRNAs and treating cardiac diseases. For example, Kang et al. used a human peripheral blood-derived exosome to load miR-21 inhibitor to reduce fibrosis in a mouse MI model [115]. However, in his study, an intra-myocardial injection of exosomes was used to increase delivery efficacy by preventing non-specific delivery to other organs, which limits its clinical application. In 2016, Li et al. designed a star-like PGMA-based nanovector (s-PGEA) with four arms that was proposed to deliver miR-29b. After intravenous injection, these s-PGEA/miR-29b complexes could transport miR-29b into heart tissues and significantly downregulate the expression of its target genes, leading to inhibition in cardiac fibrosis [94]. In addition, s-PGEA showed no toxic effects on organs, thus opening a way for further clinical application.

## 5. Conclusions and Future Prospects

Although cardiac fibrosis has been demonstrated to take part in almost all types of heart disease, the mechanisms of occurrence and development of this disease are still not fully understood. The core signaling pathway regulating cardiac fibrosis has been clarified based on experiments designed and performed in cell-culture systems and animal models.

The discovery of the first miRNA, lin-4, in 1993 in *Caenorhabditis elegans* has revolutionized the field of molecular biology [116]. Over the past 2–3 decades, miRNAs have become well known as important epigenetic regulators in the progression of cardiac fibrosis. They can directly regulate the transcription and translation of cardiac fibrosis-related genes by binding to them or influencing their expression levels through interacting with other non-coding RNAs. In this review, we comprehensively summarized the pathophysiology of cardiac fibrosis, the effects of miRNAs on cardiac fibrosis-related signaling pathways, and the potential clinical application of miRNAs in cardiac fibrosis. A growing body of evidence suggests that miRNAs regulate cardiac fibrosis and exert the role of therapeutic targets. Additionally, they can be used as diagnostic and prognostic biomarkers. Therefore, novel insights into the biological mechanisms of novel miRNAs involved in regulating cardiac fibrosis may further promote the therapy of cardiac fibrosis.

To date, some miRNAs have been found in lower organisms, such as zebrafish and newts, which have a strong capacity for cardiac regeneration and can promote cardiac regeneration post-ischemic injury [117]. For instance, Garreta et al. summarized the miRNAs controlling post-natal cardiomyocyte proliferation in non-human vertebrates such as zebrafish, neonatal and adult mice, and adult rats [118]. Manipulating these miRNAs as a novel approach to awaken the dormant regenerative potential of the adult mammalian heart may provide a new strategy for the treatment of cardiac injury and fibrosis. Future research should apply genome-editing technologies (e.g., CRISPR/Cas9 system) to further advance our understanding of the functions of miRNAs in cardiac fibrosis.

## Figures and Tables

**Figure 1 genes-13-01390-f001:**
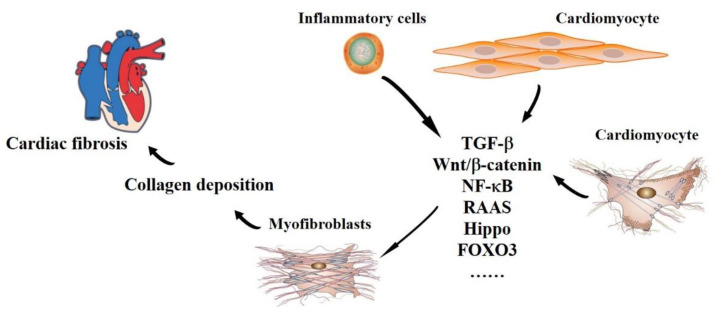
**The pathophysiology of cardiac fibrosis.** Cardiomyocytes, cardiomyocytes, and inflammatory cells participate in the process of cardiac fibrosis during related pathways.

**Figure 2 genes-13-01390-f002:**
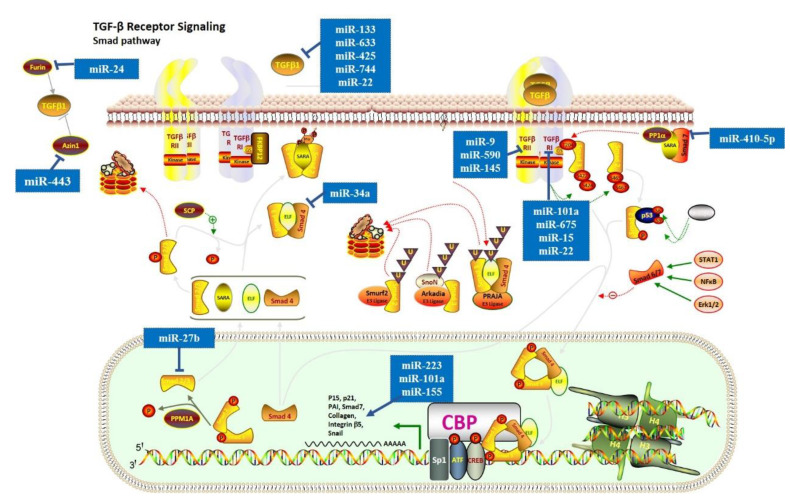
**MiRNAs in the regulation of Smad-dependent (canonical) signaling pathway.** In the canonical TGF-β signaling pathway, TGF-β1 interacts with TGFβRI/TGFβRII, leading to the phosphorylation of Smad2/3. This figure shows miRNAs that regulate the expression of the Smad-dependent (canonical) signaling pathway. The symbol of “— —>” means positive effect, and the symbol of “— —|” means negative effect.

**Figure 3 genes-13-01390-f003:**
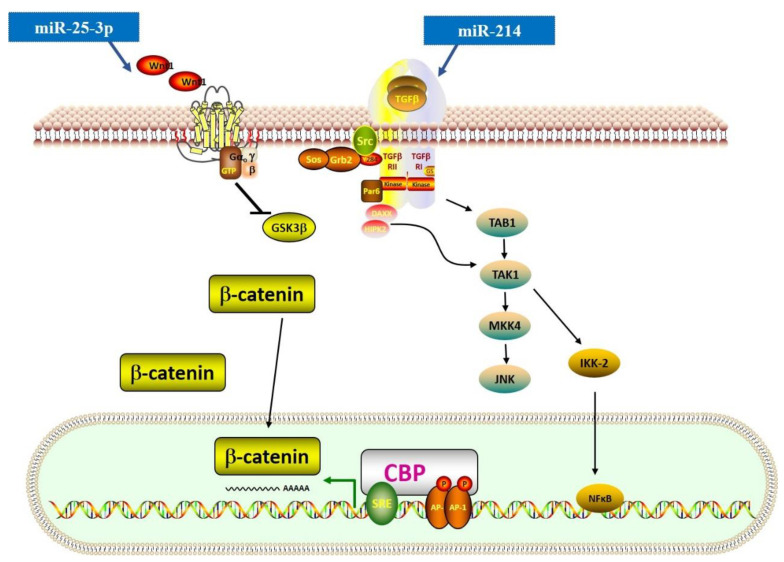
**MiRNAs in the regulation of TGF-β-related Wnt and NF-κB signaling pathways.** This figure shows miRNAs that regulate the expression of TGF-β-related Wnt and NF-κB signaling pathways. The symbol of “— —>” means positive effect, and the symbol of “— —|” means negative effect.

**Figure 4 genes-13-01390-f004:**
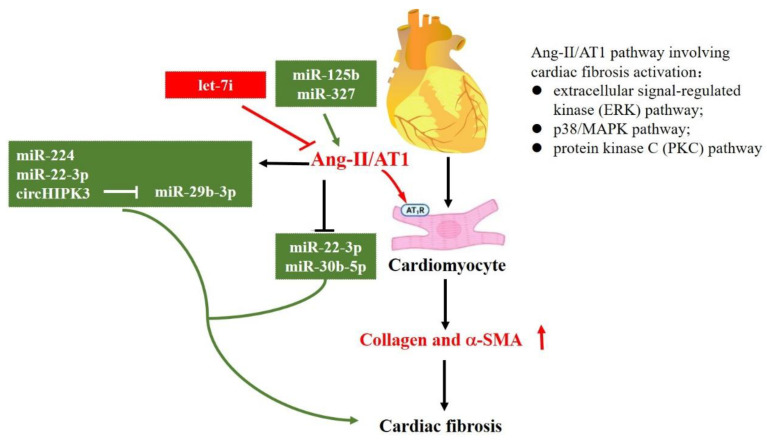
**MiRNAs in the regulation of Renin-Angiotensin-Aldosterone System (RAAS).** This figure shows miRNAs that regulate the expression of RAAS pathways. The symbol of “— —>” means positive effect, and the symbol of “— —|” means negative effect.

**Table 1 genes-13-01390-t001:** MiRNAs associated with cardiac fibrosis.

Role in CF	Name of miRNA	References
**Anti-fibrotic**		
	miR-1	[22]
	miR-101	[23,24]
	miR-133	[25,26,27]
	miR-145	[28]
	miR-15b-5p	[29]
	miR-18	[30]
	miR-214-3p	[31,32]
	miR-22	[33,34,35]
	miR-24	[36]
	miR-26a/b	[37,38]
	miR-27a/b	[39,40]
	miR-29a/b/c	[41,42,43,44]
	miR-30	[35,45,46]
	miR-384-5p	[47]
	miR-425	[48]
	miR-486a-5p	[49]
	miR-590	[25]
	miR-663	[50]
	miR-675	[51]
	miR-744	[48]
	miR-9	[52]
**Pro-fibrotic**		
	miR-10a	[53]
	miR-125	[54]
	miR-132	[55]
	miR-155	[56]
	miR-20a-5p	[55]
	miR-21	[57,58,59,60,61]
	miR-223	[62]
	miR-224	[58]
	miR-25-3p	[63]
	miR-323a-3p	[64]
	miR-327	[65]
	miR-328	[66]
	miR-34a/b/c	[67,68]
	miR-410-5p	[69]
	miR-433	[70]
	miR-4454	[71]

**Table 2 genes-13-01390-t002:** Potential therapeutic targets of miRNAs in cardiac fibrosis.

Name of miRNA	Role in CF	References
Anti-fibrotic miRNAs		
miR-29a/b/c	alleviated the ECM remodeling; reduced the expression of COL1A1-3, FBN-1, and ELN1	[43,44,92,93,94]
miR-133	improved myocardial fibrosis induced by TAC; reduced the expression of RHOA, CDC42, aNelf-A/WHSC2, and CTGF	[26,27,95]
miR-18a	prevented adult cardiac fibroblast differentiation	[30,45]
miR-1	improved function in TAC-induced cardiac remodeling and heart failure	[96]
Pro-fibrotic miRNAs		
miR-21	caused the upregulation of collagens and TGF-β canonical and non-canonical pathways MiR-21 silencing could significantly reduce cardiac fibrosis by activation of ECM degradation.	[61]
miR-34a/b/c	Inhibiting the expression of miR-34 family could prevent pressure overload-induced left ventricle remodeling and improve cardiac fibrosis.	[97]

## Data Availability

Not Applicable.

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
