# Peer review of "New Insights into the Functions of MicroRNAs in Cardiac Fibrosis: From Mechanisms to Therapeutic Strategies"

_genes, 2022, doi:10.3390/genes13081390_

Round 1
Reviewer 1 Report
Summary
The review by Zhao et al. describes signaling pathways of cardiac fibrosis and how microRNAs can regulate these pathways. The authors discuss potential therapeutic strategies for cardiac fibrosis through the modulation of microRNAs.
Major comments
- The manuscript would benefit from an early and clear definition of the different forms of cardiac fibrosis, and a description of the different pathological contexts in which they predominate. For example: Upon systemic pressure overload, fibrosis in the heart might be predominantly diffuse and reactive to the mechanical changes that are occurring. With acute injury such as myocardial infarction, fibrosis might initially have reparative functions and lead to a focal scar that replaces dying cardiomyocytes. What are some of the key differences in pro-fibrotic players, histological manifestation, and consequences for the heart in reactive versus replacement fibrosis? Recent guidelines of the ESC describe very nicely the importance of a better definition, quantification, and treatment of fibrosis in heart failure (doi: 10.1002/ejhf.1406) and might aid the authors in developing the story line.
In every chapter of the manuscript thereafter, clarity could be increased by giving the reader context which form(s) of fibrosis, and which cardiac condition(s) are being addressed.
- A comprehensive discussion on if, when, and in which contexts it might be therapeutically beneficial for the patient to block fibrosis in the heart would increase the quality of the manuscript. In line, it would be interesting to discuss potential effects of treating fibrosis without treating the underlying problem (for example pressure overload). Potential negative and potential positive consequences of limiting fibrosis in different cardiac conditions should be more thoroughly addressed.
- Line 60 - What do the authors mean by “physiological” development of fibrosis? Is that an official term and if yes, in which setting does physiological fibrosis occur?
- Line 84 – The authors are stating that apoptosis of myofibroblasts may cause a scar formation upon injury. They probably mean apoptosis of cardiomyocytes.
- It would be helpful to add a couple of sentences below the headings of the figures to describe in more detail what is shown in the figures.
- A figure showing the mode of action of microRNAs would be helpful. This figure could include therapeutic options to modulate them (inhibit versus replenish).
- Line 198 – The authors open the sentence “As illustrated in Figure 4,”. In Figure 4 the relevant cellular factors/signaling pathways still need to be added.
- Figure 4 shows a cardiomyocyte with a receptor for Ang-II. Do cardiac fibroblasts also possess Ang-II receptors?
- Chapter 3.1. TGF-β signaling pathways could benefit from the following considerations:
1) One sentence describing the impact on fibrotic behaviors of cardiac fibroblasts induced by the Smad-dependent versus Smad-independent pathway as early as line 124 would increase clarity.
2) In addition to being pro-fibrotic, TGF-β plays a role in a large number of pathways involved in cardiac repair and has even been shown to be cardioprotective and facilitate cardiac regeneration (https://doi.org/10.3389/fcvm.2019.00140). The authors should address the potential deleterious effect of limiting TGF-β in the discussion of potentially therapeutic drugs.
3) The authors describe miR-34a as a miRNA that directly targets the downstream player Smad4. Smad4 plays a pivotal role for the contractile function of cardiomyocytes (https://www.ncbi.nlm.nih.gov/pmc/articles/PMC6390466/). This raises the question how cell-type specific microRNAs are expressed, and/or if the delivery of therapeutics can be optimized in such a way that they exclusively target the cardiac fibroblast.
- The authors describe cardiac fibroblasts as one of the largest cell populations of the mammalian heart. This is controversial. Depending on the methods applied, the results in literature vary substantially.
- The role of autophagy in heart failure also seems controversial. Maybe this is due to the fact that some studies determine autophagic flux, while others quantify autophagic markers without assessing autophagic flux. Alternatively, there might be timing or amount related aspects to it. A more comprehensive discussion would be warranted.
- It would be very important to discuss general aspects on available therapeutic strategies that either inhibit or replenish microRNAs. These include:
1) Which obstacles need to be overcome to bring microRNA modulators into the clinics?
2) Which side-effects/off target effects might be anticipated?
3) Can microRNA modulating agents be made cardiac fibroblast specific?
4) Do microRNA interfering drugs go preferentially into liver and kidney? How feasible is it to target the heart specifically?
Minor comments
- The citation list seems to miss citations that appear in the table. For example, the citation list ends with ref. # 104 but in the table, but there are citations up to # 146.
- Citations should be inserted after the following sentences:
Line 48 - In addition, several types of cells, such as immune cells, cardiomyocytes, and vascular cells, can also acquire a fibro genic phenotype under certain conditions.
Line 227 - Furthermore, miR-132-3p and miR-20a-5p were also confirmed to regulate autophagy in heart failure by targeting pro-autophagic transcription factor (FOXO3a) and ATG7/SOD2, respectively.
Line 300 - Besides, miR-29, miR-133, miR-30, and miR-1 have been widely reported as the therapeutic targets of cardiac fibrosis.
Author Response
Reviewer 1
Summary
The review by Zhao et al. describes signaling pathways of cardiac fibrosis and how microRNAs can regulate these pathways. The authors discuss potential therapeutic strategies for cardiac fibrosis through the modulation of microRNAs.
Major comments
- The manuscript would benefit from an early and clear definition of the different forms of cardiac fibrosis, and a description of the different pathological contexts in which they predominate. For example: Upon systemic pressure overload, fibrosis in the heart might be predominantly diffuse and reactive to the mechanical changes that are occurring. With acute injury such as myocardial infarction, fibrosis might initially have reparative functions and lead to a focal scar that replaces dying cardiomyocytes. What are some of the key differences in pro-fibrotic players, histological manifestation, and consequences for the heart in reactive versus replacement fibrosis? Recent guidelines of the ESC describe very nicely the importance of a better definition, quantification, and treatment of fibrosis in heart failure (doi: 10.1002/ejhf.1406) and might aid the authors in developing the story line. In every chapter of the manuscript thereafter, clarity could be increased by giving the reader context which form(s) of fibrosis, and which cardiac condition(s) are being addressed.
Response: Thank you for your comment. We revised the manuscript according to the reviewer's suggestions.
- A comprehensive discussion on if, when, and in which contexts it might be therapeutically beneficial for the patient to block fibrosis in the heart would increase the quality of the manuscript. In line, it would be interesting to discuss potential effects of treating fibrosis without treating the underlying problem (for example pressure overload). Potential negative and potential positive consequences of limiting fibrosis in different cardiac conditions should be more thoroughly addressed.
Response: Thank you for your comment. We revised the manuscript according to the reviewer's suggestions.
- Line 60 - What do the authors mean by “physiological” development of fibrosis? Is that an official term and if yes, in which setting does physiological fibrosis occur?
Response: Thank you for your comment. Accordingly, we deleted the description in the revised manuscript.
- Line 84 – The authors are stating that apoptosis of myofibroblasts may cause a scar formation upon injury. They probably mean apoptosis of cardiomyocytes.
Response: Thank you for your comment. This part was revised.
- It would be helpful to add a couple of sentences below the headings of the figures to describe in more detail what is shown in the figures.
Response: Thank you for your comment. Accordingly, we added some sentences below the headings of the figures. Please see the figure legends in the revised manuscript (line 786-803).
- A figure showing the mode of action of microRNAs would be helpful. This figure could include therapeutic options to modulate them (inhibit versus replenish).
Response: Thank you for your comment. In fact, we used a table (Table 2) to summarize the mode of action of microRNAs. Please see Table 2.
- Line 198 – The authors open the sentence “As illustrated in Figure 4,”. In Figure 4 the relevant cellular factors/signaling pathways still need to be added.
Response: Thank you for your comment. As suggested, we added the relevant cellular factors/signaling pathways to Figure 4. Please see the file named “Figure 4-new.jpg”.
- Figure 4 shows a cardiomyocyte with a receptor for Ang-II. Do cardiac fibroblasts also possess Ang-II receptors?
Response: Thank you for your comment. As reported in the literature, cardiac fibroblasts also possess Ang-II receptors (Kawano H, Do YS, Kawano Y, Starnes V, Barr M, Law RE, Hsueh WA. Angiotensin II has multiple profibrotic effects in human cardiac fibroblasts. Circulation. 2000 Mar 14;101(10):1130-7.). Therefore, this part was added to the revised manuscript.
- Chapter 3.1. TGF-β signaling pathways could benefit from the following considerations:
1) One sentence describing the impact on fibrotic behaviors of cardiac fibroblasts induced by the Smad-dependent versus Smad-independent pathway as early as line 124 would increase clarity.
Response: Thank you for your comment. Accordingly, we added the sentences in the revised manuscript. Please see line 144-151 in the revised manuscript.
2) In addition to being pro-fibrotic, TGF-β plays a role in a large number of pathways involved in cardiac repair and has even been shown to be cardioprotective and facilitate cardiac regeneration (https://doi.org/10.3389/fcvm.2019.00140). The authors should address the potential deleterious effect of limiting TGF-β in the discussion of potentially therapeutic drugs.
Response: Thank you for your comment. As suggested, we added the discussion in the revised manuscript. Please see line 197-204.
3) The authors describe miR-34a as a miRNA that directly targets the downstream player Smad4. Smad4 plays a pivotal role for the contractile function of cardiomyocytes (https://www.ncbi.nlm.nih.gov/pmc/articles/PMC6390466/). This raises the question how cell-type specific microRNAs are expressed, and/or if the delivery of therapeutics can be optimized in such a way that they exclusively target the cardiac fibroblast.
Response: Thank you for your comment. In fact, we discussed the delivery of therapeutics of miR-34a in section 4.1.2. Please see the revised file.
- The authors describe cardiac fibroblasts as one of the largest cell populations of the mammalian heart. This is controversial. Depending on the methods applied, the results in literature vary substantially.
Response: Thank you for your comment. Accordingly, we deleted the description in the revised manuscript.
- The role of autophagy in heart failure also seems controversial. Maybe this is due to the fact that some studies determine autophagic flux, while others quantify autophagic markers without assessing autophagic flux. Alternatively, there might be timing or amount related aspects to it. A more comprehensive discussion would be warranted.
Response: Thank you for your comment. As suggested, we added some discussion about the controversy about autophagy in heart failure. Please see the sentences between line 270-275.
- It would be very important to discuss general aspects on available therapeutic strategies that either inhibit or replenish microRNAs. These include:
1) Which obstacles need to be overcome to bring microRNA modulators into the clinics?
2) Which side-effects/off target effects might be anticipated?
3) Can microRNA modulating agents be made cardiac fibroblast specific?
4) Do microRNA interfering drugs go preferentially into liver and kidney? How feasible is it to target the heart specifically?
Response: Thank you for your comment. As suggested, we added a section titled “4.3 New progress in delivery approaches of miRNAs” in the revised manuscript to discuss these issues. In the section, we discussed the new delivery approaches, specificity, and main problems that could affect miRNAs in clinic treatment. Please see the new section (line 412-431).
Minor comments
- The citation list seems to miss citations that appear in the table. For example, the citation list ends with ref. # 104 but in the table, but there are citations up to # 146.
Response: Thank you for your comment. As suggested, we re-organized Table 1. Please see the file named “Tables-new. docx”.
- Citations should be inserted after the following sentences:
Line 48 - In addition, several types of cells, such as immune cells, cardiomyocytes, and vascular cells, can also acquire a fibro genic phenotype under certain conditions.
Line 227 - Furthermore, miR-132-3p and miR-20a-5p were also confirmed to regulate autophagy in heart failure by targeting pro-autophagic transcription factor (FOXO3a) and ATG7/SOD2, respectively.
Line 300 - Besides, miR-29, miR-133, miR-30, and miR-1 have been widely reported as the therapeutic targets of cardiac fibrosis.
Response: Thank you for your comment. Accordingly, we inserted the citations after these sentences above. Please see the revised file.

Reviewer 2 Report
This is a timely review to explore additional modalities for designing newer generation diagnostic and therapeutic approaches for myocardial infarction. The authors sum up the various miRNAs involved in key fibrotic pathways during myocardial injury. While the list of miRNAs is comprehensive, a little more detail than just listing them in relation to the pathway would be more worthwhile. Synthesizing the data available from the literature to compile a deeper understanding of how these miRNAs actually target the pathway and affect the subsequent phenotype (rather than just saying miRNA xyz affects abc mediator) would be more useful for the target audience. Further, immune pathways are also a major driver of cardiac remodeling and fibrosis post cardiac injury and the authors touch upon this pathway only in passing.
A small section on whether there are different classes of miRNAs present in species that show cardiac regeneration (such as zebrafish) and how these influence the process and compare to those in non-regenerating pro-fibrotic species would add an interesting perspective to this review.
Additionally, while the authors mention potential therapeutic uses of miRNAs, it would help if they could also talk about newer delivery approaches that could be used. miRNAs were discovered more than a decade ago but there hasn't been a significant increase in their use as therapeutics for any disease. The authors could discuss what studies have been done (if so) clinically or are there any relevant trials in the clinic. A section on the cons or problems of using miRNA as treatment would also be of interest to the readers.
In sum, I am supportive of the publication of the review, provided the authors can flesh out the sections a little more.
Author Response
Reviewer 2
This is a timely review to explore additional modalities for designing newer generation diagnostic and therapeutic approaches for myocardial infarction. The authors sum up the various miRNAs involved in key fibrotic pathways during myocardial injury. While the list of miRNAs is comprehensive, a little more detail than just listing them in relation to the pathway would be more worthwhile. Synthesizing the data available from the literature to compile a deeper understanding of how these miRNAs actually target the pathway and affect the subsequent phenotype (rather than just saying miRNA xyz affects abc mediator) would be more useful for the target audience. Further, immune pathways are also a major driver of cardiac remodeling and fibrosis post cardiac injury and the authors touch upon this pathway only in passing.
Response: Thank you for your comment. Reviewer 2 suggested adding more detail in Table 2 than just listing them in relation to the pathway. In fact, in order to make the table concise, we only listed them in relation to the pathway, such as other Review research. We summarized it all in detail in the manuscript. As suggested, we also discussed how these miRNAs target the pathway and affect the subsequent phenotype in the revised manuscript. Please see the revised file.
A small section on whether there are different classes of miRNAs present in species that show cardiac regeneration (such as zebrafish) and how these influence the process and compare to those in non-regenerating pro-fibrotic species would add an interesting perspective to this review.
Response: Thank you for your comment. As suggested, we discussed how these cardiac regeneration-related miRNAs influence the process and then compared them to those in non-regenerating pro-fibrotic species in the section on conclusions and prospects. Please see the contents in the revised manuscript (line 452-456).
Additionally, while the authors mention potential therapeutic uses of miRNAs, it would help if they could also talk about newer delivery approaches that could be used. miRNAs were discovered more than a decade ago but there hasn't been a significant increase in their use as therapeutics for any disease. The authors could discuss what studies have been done (if so) clinically or are there any relevant trials in the clinic. A section on the cons or problems of using miRNA as treatment would also be of interest to the readers.
Response: Thank you for your comment. As suggested, we added a section titled “4.3 New progress in delivery approaches of miRNAs” in the revised manuscript. In the section, we discussed the new delivery approaches that could be used to transport miRNAs in clinic treatment. Please see the new section (line 412-431).
In sum, I am supportive of the publication of the review, provided the authors can flesh out the sections a little more.
Reviewer 3 Report
Zhao et al. reviewed the roles of MicroRNAs in Cardiac Fibrosis and their implications for therapeutic strategies. Excess ECM deposition causes cardiac fibrosis, which is linked to the majority of heart disorders and is a major public health concern worldwide. The miRNAs were discovered to be linked to the advancement of cardiac fibrosis. The authors explored the potential clinical use of miRNAs in cardiac fibrosis, as well as prospective miRNAs and miRNA-related regulatory mechanisms.
Table 1. Sort the table content "Role in CF" and "Name of the miRs" - I recommend sorting both ways, first Role in CF (anti-fibrotic and pro-fibrotic), then Name of the miRs (according to number).
Figure 2 is confusing with repetitive colouring; it is suggested that colours be changed to read the molecular name and only miRs be bold type.
Figure 3. Although miR-25-3p and miR-214 target Dickkopf (DKK) and CTRP9, respectively, those two miRs control TGF-related Wnt and NF-B signalling, this figure appears to be unrelated to that miRsmiRs and should be redrawn appropriately.
Figure 4 depicts the role of miRNAs in the regulation of the Renin-Angiotensin-Aldosterone System (RAAS). Figure confuses which regulates AngII positively and negatively, despite the regulation and relevant research provided in the text section (Maybe two different colour needs to be given for easy identification). Include literature on miRs influencing Renin and Aldosterone production enzymes to explain why the authors only considered AngII in figure4.
To increase the readability of the manuscript, care must be made to brevity, spelling, and grammar throughout the document.
Author Response
Reviewer 3
Zhao et al. reviewed the roles of MicroRNAs in Cardiac Fibrosis and their implications for therapeutic strategies. Excess ECM deposition causes cardiac fibrosis, which is linked to the majority of heart disorders and is a major public health concern worldwide. The miRNAs were discovered to be linked to the advancement of cardiac fibrosis. The authors explored the potential clinical use of miRNAs in cardiac fibrosis, as well as prospective miRNAs and miRNA-related regulatory mechanisms.
Table 1. Sort the table content "Role in CF" and "Name of the miRs" - I recommend sorting both ways, first Role in CF (anti-fibrotic and pro-fibrotic), then Name of the miRs (according to number).
Response: Thank you for your comment. As suggested, we re-organized Table 1. Please see the file named “Tables-new. docx”.
Figure 2 is confusing with repetitive colouring; it is suggested that colours be changed to read the molecular name and only miRs be bold type.
Response: Thank you for your comment. As suggested, we simplified Figure 2, and marked the miRNAs in Figure 2 and 3. Please see the file named “Figure 2-new.jpg” and “Figure 3-new.jpg”.
Figure 3. Although miR-25-3p and miR-214 target Dickkopf (DKK) and CTRP9, respectively, those two miRs control TGF-related Wnt and NF-B signalling, this figure appears to be unrelated to that miRsmiRs and should be redrawn appropriately.
Response: Thank you for your comment. As suggested, we redrew Figure 3 to make it more appropriate. Please see the file named “Figure 3-new.jpg”.
Figure 4 depicts the role of miRNAs in the regulation of the Renin-Angiotensin-Aldosterone System (RAAS). Figure confuses which regulates AngII positively and negatively, despite the regulation and relevant research provided in the text section (Maybe two different colour needs to be given for easy identification). Include literature on miRs influencing Renin and Aldosterone production enzymes to explain why the authors only considered AngII in figure 4.
Response: Thank you for your comment. As suggested, we redrew Figure 4. Please see the file named “Figure 4-new.jpg”. In addition, we mentioned that “Except for TGF-β and TGF-β related signaling pathways, the RAAS is another important signaling pathway in cardiac fibrosis. Among them, Angiotensin II (Ang-II) was reported as the most predominant isoform[6].” in the manuscript to explain why the authors only considered AngII in Figure 4. Please see line 234-236.
To increase the readability of the manuscript, care must be made to brevity, spelling, and grammar throughout the document.
Response: Thank you for your comment. A professional English proofreading company revised the manuscript. The Certificate of English Editing has been attached.
--End

Round 2
Reviewer 2 Report
The authors have addressed my questions.